# Bone Health and Endocrine Therapy with Ovarian Function Suppression in Premenopausal Early Breast Cancer: A Real-Life Monocenter Experience with Denosumab

**DOI:** 10.3390/curroncol32080421

**Published:** 2025-07-26

**Authors:** Angelachiara Rotondi, Valentina Frescura, Giorgia Arcuri, Giovanna Garufi, Letizia Pontolillo, Luca Mastrantoni, Elena Di Monte, Noemi Maliziola, Maria Antonia Fucile, Francesca Salvatori, Rita Mondello, Ilaria Poli, Gaia Rachele Oliva, Ginevra Mongelli, Antonella Palazzo, Alessandra Fabi, Emilio Bria, Giampaolo Tortora, Armando Orlandi

**Affiliations:** 1Comprehensive Cancer Center, Fondazione Policlinico Universitario Agostino Gemelli IRCCS, 00168 Rome, Italy; angelachiara.rotondi@guest.policlinicogemelli.it (A.R.); valentina.frescura@guest.policlinicogemelli.it (V.F.); giorgia.arcuri@guest.policlinicogemelli.it (G.A.); giovanna.garufi@guest.policlinicogemelli.it (G.G.); letizia.pontolillo@guest.policlinicogemelli.it (L.P.); luca.mastrantoni@guest.policlinicogemelli.it (L.M.); elena.dimonte@guest.policlinicogemelli.it (E.D.M.); noemi.maliziola@guest.policlinicogemelli.it (N.M.); mariaantonia.fucile@guest.policlinicogemelli.it (M.A.F.); francesca.salvatori@guest.policlinicogemelli.it (F.S.); rita.mondello@guest.policlinicogemelli.it (R.M.); ilaria.poli@guest.policlinicogemelli.it (I.P.); gaiarachele.oliva@guest.policlinicogemelli.it (G.R.O.); ginevra.mongelli@guest.policlinicogemelli.it (G.M.); antonella.palazzo@policlinicogemelli.it (A.P.); alessandra.fabi@policlinicogemelli.it (A.F.); giampaolo.tortora@policlinicogemelli.it (G.T.); 2Dipartimento di Medicina e Chirurgia Traslazionale, Università Cattolica del Sacro Cuore, 00168 Rome, Italy; 3Precision Medicine in Breast Cancer Unit, Department of Woman and Child Health and Public Health, Fondazione Policlinico Universitario Agostino Gemelli IRCCS, 00168 Rome, Italy; 4Medical Oncology Unit, Ospedale Isola Tiberina-Gemelli Isola, 00186 Rome, Italy

**Keywords:** early breast cancer, adjuvant endocrine therapy, premenopausal, bone health, denosumab, osteoporosis, osteonecrosis of the jaw

## Abstract

Endocrine therapy significantly reduces the recurrence risk of luminal breast cancer but also leads to decreased bone mineral density and increased osteoporotic fracture risk. This is particularly evident in premenopausal patients receiving ovarian function suppression, as abrupt estrogen deprivation disrupts bone remodeling, enhancing osteoclastic activity. ESMO guidelines recommend bisphosphonates or denosumab to prevent or treat osteopenia and osteoporosis in oncology patients. Evidence on the long-term use and discontinuation effects of denosumab in premenopausal women remains limited; at the same time, optimal initiation timing and monitoring strategies are undefined. We report data from our Comprehensive Cancer Center on denosumab in this setting, aiming to assess not only efficacy but also potential risks. Our results demonstrated clinically significant improvements in all bone health parameters, highlighting the importance of careful management in this population, which is highly vulnerable to the negative effects of acute estrogen deprivation.

## 1. Introduction

Breast cancer (BC) is the most common tumor among women [1]. In Italy alone, an estimated 55,000 new cases of BC were recorded in 2020 [2].

Early-stage diagnosis is the most frequent, and in such cases, treatment is typically multimodal, comprising surgery, radiotherapy (RT), and systemic oncologic therapies. Prognosis depends mainly on tumor stage and biological subtype at the time of diagnosis. Hormone receptor-positive/human epidermal growth factor receptor 2-negative (HR+/HER2−) disease is the most prevalent subtype of early breast cancer (eBC), representing more than 70% of cases. For patients with high-risk HR+/HER2− eBC, endocrine therapy (ET) is recommended following surgery and, when indicated, chemotherapy (ChT) and RT [3]. In premenopausal patients, tamoxifen monotherapy is indicated for low-risk tumors (e.g., stage I luminal A-like eBC). In contrast, high-risk eBC requires ovarian function suppression (OFS) with an agonist of luteinizing hormone-releasing hormone (aLHRH), in combination with either a selective estrogen receptor modulator (SERM) such as tamoxifen or an aromatase inhibitor [3]. Combination therapy with OFS plus either a SERM or an aromatase inhibitor significantly reduces the risk of locoregional and distant recurrence, as well as contralateral BC, while improving overall survival (OS) [4,5].

Although the standard duration of ET is 5 years, prolonged therapy for up to 7–10 years may offer additional benefits in terms of reducing the risk of recurrence and improving survival in high-risk patients [6,7]. However, ET is not devoid of adverse effects. Estrogen deprivation alters bone metabolism and homeostasis, causing bone loss [8,9,10]. Bone is a dynamic tissue undergoing continuous remodeling by basic multicellular units (BMUs), composed of osteoclasts, responsible for bone resorption, and osteoblasts, responsible for bone formation. With estradiol deficiency, this balance shifts toward resorption, resulting in a reduction in bone mineral density (BMD) and an increased risk of osteoporotic fractures [11,12,13,14].

Osteoporotic fractures have a significant impact on patient morbidity, quality of life (QoL), and healthcare costs [15]. Furthermore, ET in premenopausal patients induces a sudden drop in estrogen levels, in contrast to the gradual hormonal decline observed in natural menopause. This abrupt hormonal suppression confers a higher risk of bone damage compared to postmenopausal patients receiving the same therapy. The highest fracture risk was observed in patients treated with OFS in combination with an aromatase inhibitor, compared with aromatase inhibitor monotherapy or OFS plus tamoxifen [16]. Antiresorptive agents such as bisphosphonates or denosumab may reduce fracture risk in this population [17]. Current ESMO guidelines recommend bisphosphonates for the prevention of ET-induced bone loss, regardless of tumor risk category or specific bisphosphonate regimen [18]. Evidence supporting the use of adjuvant denosumab is less consistent, and data on its efficacy and safety in premenopausal patients are limited [19]. Both bisphosphonates and denosumab have been shown to increase BMD and reduce fracture risk when compared to placebo or no treatment [20]. In Italy, the prescription of antiresorptive agents is regulated by the AIFA Note 79, which allows for the use of bisphosphonates and denosumab to inhibit osteoclastic activity, increase BMD, and reduce fracture risk [21].

Denosumab, a fully human monoclonal antibody against RANK ligand (RANKL), has demonstrated clinical efficacy in improving BMD in various settings, including postmenopausal osteoporosis [22]. In eBC patients with low bone mass undergoing ET, denosumab administered biannually significantly increased trabecular and cortical BMD over 24 mo, with a safety profile comparable to placebo [23].

Given these benefits, denosumab may be a valuable strategy to reduce ET-related skeletal complications in clinical practice [8,24]. Although antiresorptive therapies are well established in postmenopausal patients, further evidence is needed to define optimal timing, duration, agent choice, and monitoring strategies in premenopausal patients undergoing ET [25]. Long-term effects of antiresorptive therapy discontinuation, as well as fracture risk management in this population, remain poorly understood. Clearer guidelines are needed on when to start and discontinue antiresorptive therapy, on the choice of the most appropriate type of antiresorptive drug, and on the optimal duration of this therapy [26].

To address this knowledge gap, we conducted a single-center, retrospective observational study to evaluate the efficacy and safety of denosumab in premenopausal patients with eBC undergoing ET. Real-world data on this high-risk, understudied population are currently limited. Our study aimed to characterize the protective effects of denosumab on bone health and its tolerability in this clinical context.

## 2. Materials and Methods

A retrospective observational study was conducted at the Comprehensive Cancer Center of Gemelli Hospital in Rome, between September 2018 and January 2025, to evaluate the use of denosumab in premenopausal patients with HR+ eBC. The study involved premenopausal women who received adjuvant ET in combination with denosumab.

Patients were selected based on the following inclusion criteria:Age between 18 and 55 years at the start of therapy;Diagnosis of early-stage and hormone receptor-positive BC;Confirmed premenopausal status at the time of diagnosis;Patients in therapy with adjuvant ET with OFS plus aromatase inhibitor or tamoxifen in combination with denosumab administered every 6 months (mo);Administration of denosumab at least twice;Availability and accessibility of baseline and follow-up clinical and laboratory data, such as BMD assessed by dual-energy X-ray absorptiometry (DEXA) or bone turnover markers as serum C-terminal telopeptide of type I collagen (CTX);Minimum follow-up of at least 12 mo.Exclusion criteria were as follows:Age over 55 years at the start of therapy;Adjuvant ET with aromatase inhibitor or tamoxifen in monotherapy, without concurrent OFS;Incomplete or unavailable clinical or imaging data for efficacy and safety analysis;Pre-existing non-oncological bone diseases (e.g., osteomalacia);Previous exposure to bisphosphonates or other antiresorptive agents prior to start of denosumab.

The primary endpoint of the study was to evaluate the efficacy, safety, and treatment adherence associated with Denosumab in this patient population.

Efficacy was assessed through changes in BMD, measured by DEXA scan, and through evaluation of bone turnover markers, particularly CTX levels. The safety profile was assessed by recording adverse events related to ET (e.g., osteoporotic fractures) and denosumab, with particular attention to osteonecrosis of the jaw (ONJ), the most serious potential complication of denosumab therapy. A descriptive analysis was performed to evaluate baseline and treatment-related characteristics, including age, smoking habit, body mass index (BMI), blood vitamin D levels and supplementation, comorbidities, tumor stage, type of surgery, type of ET, and the use of additional therapies such as RT, CDK4/6 inhibitors, anti-HER2 agents, and ChT. Treatment adherence was checked for both ET and denosumab administration during the observation period (Table 1).

The study was designated as a non-profit research study and received unanimous approval from the Ethics Committee. (Ethics Committee Name: Fondazione Policlinico Universitario Agostino Gemelli IRCCS-Università Cattolica del Sacro Cuore, Comitato Etico. Approval Code: Prot. ID 5519 Approval Date: 9 February 2023.).

All study procedures were performed in accordance with the approved protocol and Good Clinical Practice guidelines.

## 3. Results

Out of a total of 108 patients with eBC treated with denosumab, only 69 met all the predefined inclusion criteria and were included in the final analysis. The remaining 39 patients were excluded due to one or more exclusion criteria, such as incomplete clinical or imaging data, prior exposure to antiresorptive agents, or lack of concurrent OFS during ET. The final cohort of 69 patients formed the study population for the subsequent efficacy and safety evaluations (Figure 1).

### 3.1. Patients’ Characteristics

The median age was 45 years, with 88.4% of patients being younger than 50 years of age. Only 2 patients (2.9%) were active smokers, while 85.5% reported never having smoked. Most patients (75.4%) had a BMI < 25; 13% were overweight (BMI 25–29), and 5.8% were classified as obese (BMI ≥ 30). Only 6 patients presented with baseline hypovitaminosis D (<20 ng/mL); 47.8% had values within the normal range (20–40 ng/mL), and 33.3% had values > 40 ng/mL. Nearly half of the patients (43.5%) received cholecalciferol supplementation during the observation period. It was found that, of 30 patients receiving vitamin D supplementation from baseline, only 2 patients experienced a worsening of BMD at the last DEXA scan assessment, 25 patients achieved an improvement in BMD, and for 3 patients we did not have DEXA scan data available for the last assessment. Among comorbidities relevant to the analysis, coeliac disease and non-coeliac gluten sensitivity were reported in two patients.

Patients enrolled in the study were initially diagnosed with eBC at stage I (50.8%), stage II (34.8%), and stage III (7.2%) according to the International TNM Classification. ChT was administered to 63.8% of patients, and anti-HER2 therapy to 14%. Following surgery, 55% of patients underwent RT. Four patients received the CDK4/6 inhibitors, abemaciclib (Table 2).

### 3.2. Adherence to Therapy and Complications

#### 3.2.1. Endocrine Therapy

All patients received adjuvant ET: 89.9% with Exemestane plus OFS, and 10.1% with tamoxifen plus OFS. Only two patients discontinued treatment with an aLHRH following ovarian adnexectomy: one due to synchronous ovarian carcinoma, and one as prophylaxis due to a pathogenic BRCA1/2 mutation. The median duration of ET was 56 mo, with the majority (85.5%) still receiving treatment at the end of the observation period. All patients were alive at the time of analysis, with only one experiencing recurrence with liver metastases; this patient was the only recorded death at the time of observation. The cause of death in this patient was progression of BC. At the time of death, the patient had already discontinued denosumab therapy for 18 mo. Therefore, we excluded a correlation between death and denosumab therapy.

Clinical and radiological follow-up assessments were conducted every 6 mo to monitor bone health status and adherence to treatment with bone-modifying agents. No bone fractures were reported (Table 2).

#### 3.2.2. Denosumab Therapy

Overall adherence to denosumab treatment was high: a total of 89.9% of patients received therapy every 6 mo as scheduled. Reduced adherence was observed in 5.8% of patients: 1 patient discontinued denosumab after 15 mo due to dental procedures and did not resume; 1 patient discontinued after a single dose due to the development of ONJ and did not resume; and 1 patient interrupted treatment for 12 mo due to dental procedures and subsequently resumed. Data were unavailable for 4.3% of patients (Table 2).

### 3.3. Bone Density and CTX Level

During follow-up, bone turnover was assessed using serum CTX levels, and BMD was evaluated by T-score via DEXA scans, in accordance with World Health Organization (WHO) criteria. CTX levels were evaluated in 35.8% of patients before and during Denosumab treatment. Baseline CTX levels were ≤0.5 ng/mL in 23.2% of these patients, between 0.5 and 1.0 ng/mL in 5.8%, and >1.0 ng/mL in 7.2%. At the last follow-up, a reduction in CTX levels was observed in 97% of evaluated patients, with a marked decrease in the proportion with values > 0.5 ng/mL. Most patients underwent a baseline DEXA scan for BMD assessment. Sixteen patients lacked an initial evaluation. For twenty-four patients, data on the most recent DEXA assessment were not available. At baseline, spinal osteoporosis was present in 20.3% of patients, and spinal osteopenia in 39.1%. A T-score < −1.0 (indicating abnormal bone mass) was found in 17.4% of patients. At the last evaluation, a general improvement in T-score was observed, with an increased proportion of patients showing normal values and a reduction in the proportion with T-scores ≤ −2.5. Femoral osteoporosis was present in 5.8% of patients, and femoral osteopenia in 50.7% at baseline. A T-score < −1.0 was observed in 20.3% of patients. The final evaluation demonstrated an overall improvement in femoral T-scores, with an increase in the percentage of patients with normal bone density and a decrease in the prevalence of osteopenia (Figure 2).

## 4. Discussion

The results of our single-center retrospective study provide significant evidence on the efficacy and safety of denosumab in preventing ET-induced bone damage in premenopausal patients with eBC. The clinical importance of these data is particularly relevant considering that this population represents one of the most complex challenges in oncological bone health management, characterized by a peculiar vulnerability to the negative effects of acute estrogen deprivation. The estrogen deprivation induced by the combination of ovarian suppression and aromatase inhibitors leads to more severe bone damage compared to natural menopause. Our data confirm this vulnerability: at baseline, 59.4% of patients already presented alterations in BMD (20.3% osteoporosis, 39.1% vertebral osteopenia), underlining the urgency of early preventive interventions. The rapidity of bone loss in this context is well documented in the literature, with studies reporting a 2–4% annual reduction in BMD during the first years of aromatase inhibitor therapy combined with ovarian suppression, significantly higher than the 1% annual loss observed in natural menopause [11,12,13]. The metabolic impact of ET in premenopausal women extends beyond simple estrogen deficiency. The sudden hormonal suppression, in contrast to the gradual decline observed in natural menopause, creates an abrupt shift in bone remodeling balance toward resorption. This is mediated by the loss of estrogen’s protective effects on osteoblast function and the removal of its inhibitory action on osteoclast activity [9,10]. The RANK/RANKL/OPG pathway, which is central to bone homeostasis, becomes dysregulated, leading to increased bone turnover and progressive bone loss [14]. Our results demonstrate clinically significant improvement in all bone health parameters following denosumab treatment. The dramatic reduction in vertebral osteoporosis from 20.3% to 5.8% (a 71.4% relative reduction) and the substantial increase in normal BMD from 17.4% to 34.8% (a 100% increase) represent clinically meaningful benefits. These improvements are particularly noteworthy considering they were achieved despite continuation of ET, demonstrating that denosumab not only prevents bone loss but determines active recovery of BMD. Similar improvements in femoral parameters confirm the systemic efficacy of treatment, addressing both trabecular and cortical bone compartments. The stable reduction in CTX levels in 97% of evaluated patients represents an objective biomarker of denosumab’s antiresorptive efficacy. This finding is consistent with the drug’s mechanism of action as a specific RANKL inhibitor, confirming effective inhibition of osteoclast activity even in the presence of intense catabolic stimuli such as severe estrogen deprivation [22]. The biochemical response parallels the densitometric improvements, providing mechanistic validation of the observed clinical benefits. The excellent safety profile observed in our study (1.4% ONJ, 0% fractures) is clinically reassuring and consistent with literature data in oncological populations. The low incidence of ONJ is particularly significant considering the median treatment duration of 33 mo, high therapeutic adherence (89.9%), and the young population with potential long exposure [18,19]. This safety profile compares favorably with bisphosphonates, which have been associated with similar or higher rates of ONJ and additional concerns such as atypical femoral fractures in long-term use scenarios. The exceptional adherence rate of 89.9% observed in our study is superior to that typically reported for other supportive therapies in oncology, suggesting good acceptability of the bimonthly administration regimen. This aspect is crucial for long-term efficacy of bone damage prevention, as inconsistent treatment significantly reduces the protective benefits. The convenience of subcutaneous administration every 6 mo, compared to monthly intravenous bisphosphonates, likely contributes to this high adherence rate and represents a practical advantage in clinical practice. Our results confirm and expand the evidence from previous prospective studies. The ABCSG-18 trial demonstrated denosumab’s superiority over placebo in terms of BMD and fracture risk reduction in postmenopausal patients receiving adjuvant aromatase inhibitor therapy [8]. The D-CARE analysis, while not showing significant impact on primary oncological outcomes, evidenced benefits on bone health parameters [18]. Recent meta-analyses support the use of bone-modifying agents in high-risk patients, with denosumab showing superior efficacy compared to bisphosphonates in terms of BMD improvement [20]. The specificity of our study lies in its focused attention on premenopausal patients with real-world follow-up, a population often under-represented in registration studies.

The positioning of denosumab in current guidelines is evolving. The 2024 ESMO recommendations indicate antiresorptive agents for prevention of ET -induced bone damage [3]. Denosumab emerges as a preferential option in premenopausal patients due to its superior efficacy compared to bisphosphonates in terms of BMD increase, dosing convenience (bimonthly administration), favorable safety profile, and reversible action upon discontinuation. The AIFA Note 79 in Italy allows for denosumab prescription to inhibit osteoclastic activity, increase BMD, and reduce fracture risk, providing regulatory support for its clinical use [21].

Several clinical questions remain unresolved and require further research. The optimal timing of treatment initiation remains debated, with options including concurrent start with ET versus initiation guided by densitometric thresholds. The duration of treatment presents another challenge, particularly regarding correlation with ET duration (5–10 years) and safe discontinuation strategies. The rebound effect, characterized by rapid bone loss post-discontinuation, requires careful management and transition strategies to alternative therapies [27].

There is still insufficient data to make definitive recommendations.

The marked loss of bone mass during adjuvant endocrine therapy (OFS + aromatase inhibitors, but also tamoxifen, which does not protect bone in premenopausal women and acts as a partial antagonist inducing bone loss) in premenopausal patients leads to high annual rates of BMD reduction, with a substantial risk of osteoporosis and fractures. In premenopausal women, there are few studies investigating the efficacy of antiresorptive therapy in preventing ET-induced BMD loss and no studies evaluating fracture outcomes. It would be useful to have a preventive assessment of bone risk and constant monitoring of BMD [26].

Risk stratification to identify patients at very high risk who need early intervention remains an area of active investigation, with potential roles for genetic and metabolic factors.

Most guidelines suggest discontinuing treatment at the end of ET unless there is a persistent high risk of bone fractures. Since denosumab, unlike bisphosphonates, is not retained in the bone, a rapid increase in bone remodeling occurs shortly after treatment discontinuation, with a reduction in BMD and a consequent high risk of vertebral fractures, with a greater risk for patients who have had a previous vertebral fracture, either before or during treatment. We also know that some preclinical data support the idea that the bone microenvironment is important in cancer progression. In fact, increased bone remodeling appears to be associated with the release of factors that stimulate tumor growth. For this reason, good adherence to treatment is important. It is equally important to consolidate the discontinuation of denosumab with a bisphosphonate. Most studies evaluating the effects of ET on bone health are short-term effects. In particular, for premenopausal women, the long-term effects of treatment on bone health and fracture risk are still unknown [26].

Cost-effectiveness considerations strongly support the investment in bone damage prevention given the substantial economic and social burden of osteoporotic fractures. Recent European data demonstrate that fragility fractures cost European healthcare systems EUR 56.9 billion annually, with this burden projected to increase by 23% to EUR 47.4 billion by 2030 in the six largest EU countries alone [28,29]. At the individual patient level, osteoporotic fractures impose significant costs: US Medicare data show mean all-cause healthcare costs of USD 47,163 in fracture patients versus USD 16,035 in controls, with hip fractures incurring the highest costs at over USD 71,000 [30]. European studies report annual fracture-related care costs of approximately EUR 900 compared to EUR 110 for osteoporosis management [31]. The per capita cost of osteoporotic fractures varies significantly across European countries, with Switzerland reporting EUR 403, Denmark EUR 251, Sweden EUR 230, and Germany EUR 167 per capita annually [32]. These substantial direct costs do not account for indirect costs including loss of quality of life, functional autonomy, informal caregiving burden, and risk of subsequent fractures. The cost-effectiveness ratio of denosumab appears highly favorable, especially in young patients with long life expectancy, when considering the prevention of fractures and their associated substantial morbidity and healthcare costs [33,34]. The prevention of bone damage extends beyond supportive care and represents an essential element of comprehensive cancer care. Fracture risk reduction has direct implications for long-term quality of life, functional autonomy and independence, compliance with ET (reduction in interruptions due to bone toxicity), and OS outcomes through prevention of immobilization-related complications. The proactive management of bone health therefore represents a paradigm of precision medicine applied to supportive care, where personalized preventive intervention can significantly impact long-term outcomes and global patient wellbeing. We acknowledge several methodological limitations characteristic of retrospective design. Missing data represent a significant challenge, with CTX levels available in only 36.2% of patients at baseline and 79.7% missing at final follow-up. DEXA evaluations were incomplete in 23.2% without baseline and 34.8% without final follow-up assessments. The limited follow-up with a median of 33 mo is insufficient to evaluate long-term fracture outcomes. The absence of a control group makes it impossible to quantify the absolute benefit versus standard of care, though ethical considerations make placebo-controlled studies difficult in this high-risk population.

Despite these limitations, our real-world data provide valuable insights into denosumab’s effectiveness in routine clinical practice. The consistency of our results with prospective trials supports the external validity of our findings and their applicability to broader clinical populations. The integration of densitometric and biochemical parameters strengthens the evidence for denosumab’s beneficial effects on bone metabolism in this vulnerable population.

## 5. Conclusions

Our study confirms that denosumab represents an effective and safe strategy for preventing bone damage in premenopausal patients with eBC undergoing adjuvant ET. The results demonstrate clinically relevant efficacy with substantial improvement in all bone health parameters, acceptable safety with low incidence of serious adverse events, clinical feasibility with high adherence and treatment acceptability, and durable benefit with maintenance of improvements throughout the observation period.

The prevention of bone damage should be considered an integral part of the therapeutic plan for every premenopausal patient with BC for prolonged ET. Proactive management of bone health is essential, as an early integrated approach with antiresorptive therapy, vitamin D supplementation, and monitoring of BMD and bone turnover would help to safeguard bone health, improve patients’ quality of life, and limit the risk of osteoporosis and therefore bone fractures. It is therefore essential to address this issue, especially in the early stages of the disease in young premenopausal patients.

Our study results support the implementation of structured bone health management protocols that include early evaluation, timely treatment, and systematic monitoring. The ultimate goal is to ensure that the success of oncological therapy is not compromised by preventable skeletal complications, guaranteeing our patients not only prolonged survival but also optimal quality of life during and after completion of their therapeutic journey. Proactive bone health management therefore represents a paradigm of precision medicine applied to supportive care, where personalized preventive intervention can significantly impact long-term outcomes and global patient wellbeing. Future research should focus on optimizing treatment timing, duration, and discontinuation strategies through prospective comparative studies, while developing personalized risk prediction models to guide treatment decisions in this high-risk population.

## Figures and Tables

**Figure 1 curroncol-32-00421-f001:**
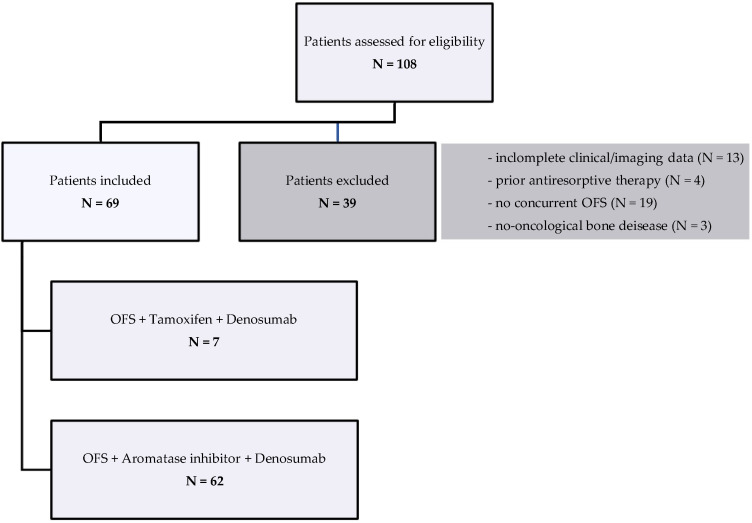
Patient selection flow chart. Of 108 patients with eBC treated with denosumab, 69 met all eligibility criteria and were included in the final analysis.

**Figure 2 curroncol-32-00421-f002:**
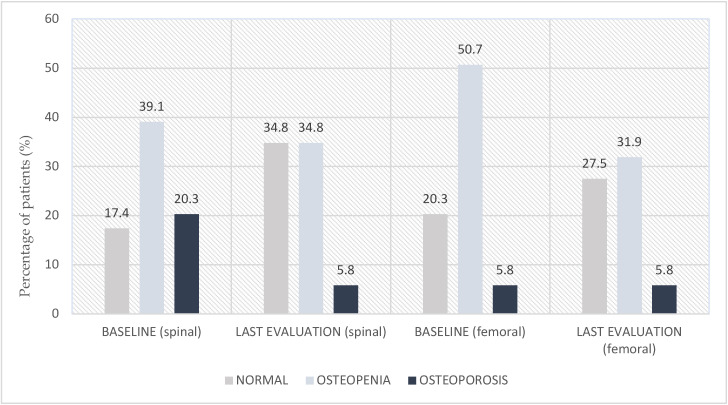
Graphical representation of the impact of denosumab therapy on BMD, assessed by DEXA scan, comparing baseline and last assessment. The T-score is the key parameter for interpreting results according to the following definitions, established by international guidelines. Normal: T-score is >−1.0; osteopenia: T-score is >−2.5 and ≤−1.0; osteoporosis: T-score is ≤−2.5.

**Table 1 curroncol-32-00421-t001:** Baseline characteristics and treatment-related variables assessed in the study.

Variable	Assessment	Description
**Demographics**		
Age	years	at baseline
Smoking habit	current/never/former	collected; baseline
BMI	kg/m^2^	at baseline
**Bone health parameters**		
BMD	DEXA scan	at baseline; during follow-up
Blood vitamin D level	ng/mL	at baseline
Vitamin D intake	yes/no	at baseline; during follow-up
**Comorbidities**	medical history	e.g., hypertension, diabetes…
**Cancer-related parameters**		
Tumor stage	TNM ^1^	at diagnosis
Type of surgery	mastectomy/quadrantectomy	-
**Systemic therapy**		
ET type	OFS + aromatase inhibitor or tamoxifen	current
CDK4/6 inhibitor use	yes/no	concomitant
Anti-HER2 agent use	yes/no	concomitant
ChT	yes/no	prior or concomitant
RT	yes/no	prior or concomitant
**Bone turnover markers**		
Serum CTX level	ng/mL	at baseline; during follow-up
**Safety**		
Bone fractures	yes/no	number and site
ONJ	yes/no	confirmed by oral surgeon
**Adherence to therapy**		
ET	adherence rate	during follow-up
Denosumab	adherence rate	during follow-up

^1^ TNM classification of malignant tumors is an international system for classifying cancers, from which the stage can be determined (TNM: Tumor, Nodes, Metastasis).

**Table 2 curroncol-32-00421-t002:** Description of patient characteristics, ET, denosumab therapy and bone health data.

	Characteristics	Patients N (%)
Patients’ characteristics		
Age	<50	61 (88.4)
	≥50	8 (11.6)
	Median age	45
	Current	2 (2.9)
Smoking habits	Former	8 (11.6)
	Never	59 (85.5)
	<25	52 (75.4)
BMI	25–29	9 (13)
	≥30	4 (5.8)
	Missing	4 (5.8)
	<20	6 (8.7)
Blood vitamin D level	20–40	33 (47.8)
	>40	23 (33.3)
	Missing	7 (10.2)
Vitamin D intake	Yes	30 (43.5)
	No	39 (56.5)
	I	35 (50.8)
Tumor stage	II	24 (34.8)
	III	5 (7.2)
	Unknown	5 (7.2)
Surgery	Quadrantectomy	43 (62.3)
	Mastectomy	26 (37.7)
ChT	Yes	44 (63.8)
	No	25 (36.2)
Anti-HER2 agents	Yes	10 (14.5)
	No	59 (85.5)
CDK4/6 inhibitors	Yes	4 (5.8)
	No	65 (94.2)
RT	Yes	38 (55)
	No	31 (45)
Recurrence	Yes	1 (1.5)
	No	68 (98.5)
Survival status		
	Alive	63 (91.3)
	Died	1 (1.5)
	Lost to follow-up	5 (7.2)
**ET**		
Type	OFS * + aromatase inhibitor	62 (89.9)
	OFS * + tamoxifen	7 (10.1)
	Ended	10 (14.5)
Time	Ongoing	59 (85.5)
	Median time	56 mo
	No	31 (45)
Bone fractures	Yes	0 (0)
	No	69 (100)
**Denosumab therapy**		
	Ended	11 (16)
Time	Ongoing	58 (84)
	Median time	33 mo
	Yes	62 (89.9)
Adherence	No	4 (5.8)
	Missing	3 (4.3)
ONJ	Yes	1 (1.4)
	No	68 (98.6)
	Missing	3 (4.3)
**BMD and bone turnover marker**		
	T-score ≤ −2.5	14 (20.3)
Spinal DEXA scan (at baseline)	T-score > −2.5 and ≤−1.0	27 (39.1)
	T-score > −1.0	12 (17.4)
	Unknown	16 (23.2)
	T-score ≤ −2.5	4 (5.8)
Femoral DEXA scan (at baseline)	T-score > −2.5 and ≤−1.0	35 (50.7)
	T-score > −1.0	14 (20.3)
	Unknown	16 (23.2)
	T-score ≤ −2.5	4 (5.8)
Spinal DEXA scan (at last follow-up)	T-score > −2.5 and ≤−1.0	24 (34.8)
	T-score > −1.0	17 (34.8)
	Unknown	24 (24.6)
	T-score ≤ −2.5	4 (5.8)
Femoral DEXA scan (at last follow-up)	T-score > −2.5 and ≤−1.0	22 (31.9)
	T-score > −1.0	19 (27.5)
	Unknown	24 (34.8)
	≤0.5	16 (23.2)
Serum CTX level (at baseline)	> 0.5 and ≤1.0	4 (5.8)
	>1.0	5 (7.2)
	Unknown	44 (63.8)
	≤0.5	10 (14.5)
Serum CTX level (at last follow-up)	>0.5 and ≤1.0	3 (4.3)
	>1.0	1 (1.5)
	Unknown	55 (79.7)

* Bilateral adnexectomy in 2.9% of patients.

## Data Availability

Data available on request due to restrictions (e.g., privacy, legal or ethical reasons).

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
