# Peer review of "Bone Health and Endocrine Therapy with Ovarian Function Suppression in Premenopausal Early Breast Cancer: A Real-Life Monocenter Experience with Denosumab"

_curroncol, 2025, doi:10.3390/curroncol32080421_

Round 1

Reviewer 1 Report

Comments and Suggestions for Authors

The results from the study entitled “ Bone health and endocrine therapy with ovarian function 2 suppression in premenopausal early breast cancer: 3 a real-life monocenter experience with denosumab” by Rotondi et al., dy demonstrate clinically significant improvement in all bone health parameters following denosumab treatment. There was a reduction in vertebral osteoporosis from 20.3% to 5.8% and increase in normal BMD, from 17.4% to 34.8% represent clinically meaningful benefits. These improvements are particularly noteworthy, considering they were achieved despite the continuation of endocrine therapy, demonstrating that denosumab not only prevents bone loss but also facilitates active recovery of bone mineral density

The authors acknowledge several limitations, including missing data for final CTX and DEXA measurements, and limited follow-up for fracture outcomes.

The focus on premenopausal patients with real-world follow-up is a strength, as this population is often underrepresented in clinical trials.

However, some points need to be addressed:

  1. One patient died during the study, please clarify the cause of death and discuss whether it was related to treatment or comorbidities.
  2. Provide further detail on the role of vitamin D supplementation. Did it contribute to the observed improvements in BMD? Were outcomes stratified based on vitamin D use?
  3. Emphasize that bone damage prevention should be considered a core component of treatment planning for all premenopausal breast cancer patients undergoing long-term endocrine therapy. Proactive bone health management can significantly affect quality of life and long-term outcomes.
  4. The recent randomized clinical trial by Ramchand et al. (2024), which examined the efficacy of denosumab in preventing bone loss in premenopausal women by suppressing bone remodeling, is referenced but not adequately discussed. This important study should be reviewed in greater depth and integrated into the discussion to support the current findings.
  5. Please label the y-axis in Figure 2 with the appropriate unit of measurement.

Author Response

  • REVISION 1.1 One patient died during the study, please clarify the cause of death and discuss whether it was related to treatment or comorbidities”.

In response to your request, we have clarified that the only patient who died in the study is the only patient who experienced a liver recurrence of disease. The cause of death was disease progression. We have added the following to section 3. Results/3.2. Adherence to therapy and complications/3.2.1. Endocrine therapy: “The cause of death in this patient was progression of breast cancer. At the time of death, the patient had already discontinued denosumab therapy for 18 months. Therefore, we excluded a correlation between death and denosumab therapy.”

  • REVISION 1.2 Provide further detail on the role of vitamin D supplementation. Did it contribute to the observed improvements in BMD? Were outcomes stratified based on vitamin D use?”

Our analysis showed that most patients who took vitamin D supplements had improved BMD on DEXA scans. However, the lack of data on patient compliance with vitamin D supplementation and the difference in vitamin D dosage between patients prevented us from stratifying the DEXA scan and CTX results according to vitamin D supplementation.

We have added the following to section 3. Results/3.1. Patients’ characteristics: In our patient population, 43.5% took vitamin D supplements. It was found that of 30 patients receiving vitamin D therapy from baseline, only 2 patients experienced a worsening of BMD at the last DEXA scan assessment, 25 patients achieved an improvement in BMD, and for 3 patients we did not have DEXA scan data available for the last assessment.”

  • REVISION 1.3 “Emphasize that bone damage prevention should be considered a core component of treatment planning for all premenopausal breast cancer patients undergoing long-term endocrine therapy. Proactive bone health management can significantly affect quality of life and long-term outcomes.”

With regard to your request, we have argued this point as follows.

We have added the following to section 5. Conclusion: “Proactive management of bone health is essential, as an early integrated approach with antiresorptive therapy, vitamin D supplementation, and monitoring of BMD and bone turnover would help to safeguard bone health, improve patients' quality of life, and limit the risk of osteoporosis and therefore bone fractures. It is therefore essential to address this issue, especially in the early stages of the disease in young premenopausal patients.

  • REVISION 1.4The recent randomized clinical trial by Ramchand et al. (2024), which examined the efficacy of denosumab in preventing bone loss in premenopausal women by suppressing bone remodeling, is referenced but not adequately discussed. This important study should be reviewed in greater depth and integrated into the discussion to support the current findings.”

We appreciated this comment and felt it was entirely appropriate to elaborate further on the above-mentioned article in our manuscript.

  • We have therefore added the following to the section 1. Introduction: “Clearer guidelines are needed on when to start and discontinue antiresorptive therapy, on the choice of the most appropriate type of antiresorptive drug, and on the optimal duration of this therapy [27].”
  • We have added the following to the section Discussion: “There is still insufficient data to make definitive recommendations. The marked loss of bone mass during adjuvant endocrine therapy (OFS + aromatase inhibitors, but also tamoxifen, which does not protect bone in premenopausal women and acts as a partial antagonist inducing bone loss) in premenopausal patients leads to high annual rates of BMD reduction, with a substantial risk of osteoporosis and fractures. In premenopausal women, there are few studies investigating the efficacy of antiresorptive therapy in preventing ET-induced BMD loss and no studies evaluating fracture outcomes. It would be useful to have a preventive assessment of bone risk and constant monitoring of BMD [27].”
  • We have added the following to the section Discussion: Most guidelines suggest discontinuing treatment at the end of ET unless there is a persistent high risk of bone fractures. Since denosumab, unlike bisphosphonates, is not retained in the bone, a rapid increase in bone remodeling occurs shortly after treatment discontinuation, with a reduction in BMD and a consequent high risk of vertebral fractures, with a greater risk for patients who have had a previous vertebral fracture, either before or during treatment. We also know that some preclinical data support the idea that the bone microenvironment is important in cancer progression. In fact, increased bone remodeling appears to be associated with the release of factors that stimulate tumor growth. For this reason, good adherence to treatment is important. It is equally important to consolidate the discontinuation of denosumab with a bisphosphonate. Most studies evaluating the effects of ET on bone health are short-term. In particular, for premenopausal women, the long-term effects of treatment on bone health and fracture risk are still unknown [27].”

  • REVISION 1.5 Please label the y-axis in Figure 2 with the appropriate unit of measurement.”
    We labeled the y-axis in Figure 2 with the appropriate unit of measurement, namely “Percentage of patients (%)”.See page 8 of 14.

Reviewer 2 Report

Comments and Suggestions for Authors

The present retrospective study was undertaken to investigate the efficacy of denosumab in improving bone density and reducing bone turnover in premenopausal patients treated for luminal breast cancer. The authors have found that, compared to endocrine therapy, denosumab resulted in decreased spinal osteoporosis and osteopenia, with the absence of fractures, together with femoral improvements. They also noted the limitations of their study (limited number of patients, limited follow-up time).

The study provides some interesting observations. However, there are several points needing correction and/or clarification, as follows.

  1. Almost every one of the references is written in a different format.
  2. To check the abbreviations used (see: eBC and EBC); to change selected abbreviations (for example, AI nowadays is used for artificial intelligence); remove the abbreviations from abstract.
  3. Approval of the bioethics committee should be provided.

Author Response

  • REVISION 2.1 “Almost every one of the references is written in a different format.”

We have corrected the “References” section: all bibliographic references are written in the same format, with the exception of references 1, 2, and 21, which contain links.

  • REVISION 2.2 To check the abbreviations used (see: eBC and EBC); to change selected abbreviations (for example, AI nowadays is used for artificial intelligence); remove the abbreviations from abstract.”

Thank you for bringing this to our attention. We have corrected the abbreviations in our manuscript as listed below:

- we have checked all abbreviations used in the text. In particular, as requested, we have corrected the abbreviation “EBC” to “eBC” and removed the abbreviation AI throughout the manuscript (previously used for “aromatase inhibitor”);

- we have removed the abbreviations in the abstract and inserted them starting from the first paragraph that follows;

- we have reviewed all abbreviations, removing any repetitions and specifying the meaning of the acronym only once (i.e., the first time the word appears in the text).

All corrections have been highlighted to facilitate your review.

  • REVISION 2.3 “Approval of the bioethics committee should be provided.”

The study was designated as a non-profit research study and received unanimous approval from the Ethics Committee. Ethics Committee Name: Fondazione Policlinico Universitario Agostino Gemelli IRCCS - Università Cattolica del Sacro Cuore, Comitato Etico. Approval Code: Prot. ID 5519 Approval Date: February 9, 2023.

All study procedures were performed in accordance with the approved protocol and Good Clinical Practice guidelines. We added as request this information in the “patients and method” section.